# Optimization Design and Experimentation of a Soil Covering Device for a Tree Planting Machine

**Xun Wu [1], Zhen Jiang [1], Lixin Zhang [1,2,*], Xue Hu [1] and Wenchun Li [1,3]**

1   School of Mechanical and Electrical Engineering, Shihezi University, Shihezi 832000, China; 20212109038@stu.shzu.edu.cn (Z.J.)
2   Energy Development Research Institute of Xinjiang Production and Construction Corps, Shihezi 832000, China
3   Tumushuke Yinfeng Modern Agricultural Equipment Co., Ltd., Tumushuke 844000, China
*   Correspondence: zhlx2001329@163.com

**Abstract:** In an effort to improve the soil covering effect of tree planting machines and reduce the lodging rate of seedlings, this paper analyzes the soil covering process and working principle of the soil covering device of tree planting machines and designs a soil covering device with adjustable soil covering width. With the help of the discrete element simulation software EDEM, the orthogonal test was designed with the width of the covering soil and the inclination angle as the test factors and the soil backfill rate as the response index. The regression analysis of the test results was carried out by Design-expert, and the optimal parameter combination of the covering soil device was determined as follows: the width of the covering soil: 600 mm, the forward angle: 45°, and the depth of the soil: 20 mm. The prototype verification test was carried out under the optimal parameters, and the soil backfill rate was 88.4%, which was basically consistent with the optimized parameters. The soil covering device with adjustable width designed in this study is of great significance to improve the soil covering rate of tree planting machines and reduce the lodging rate of seedlings.

**Keywords:** tree planting machine; soil covering device; EDEM; regression analysis; optimization design

## 1. Introduction

In recent years, with the more and more serious land desertification in southern Xinjiang, the available land resources of human beings have been reduced, and the land productivity has also declined significantly, which has seriously affected people's production and life. Therefore, the design and development of an automatic and efficient tree planting machine suitable for desert areas in southern Xinjiang and the design and study of its key components are of great significance to solve the problems of land desertification and high cost of artificial tree planting in southern Xinjiang.

As the key structural component of the tree planting machine, the soil covering device of the tree planting machine will directly affect the tree planting effect. In the process of planting trees, the factors that have the greatest impact on the survival rate and lodging rate are mainly planting tree depth, soil backfill rate, soil suppression effect, etc. The structure and parameters of the soil covering device will directly affect the soil backfill rate. In order to solve the problem of the low survival rate and high lodging rate of seedlings caused by the low soil backfilling rate of tree planting machines, it is necessary to study the design of soil-covering devices for tree planting machines.

At present, there are many kinds of soil-covering devices applied to agricultural machinery at home and abroad. According to the structural form, they are mainly divided into the following categories, namely single disc type, eight-shaped scraper type, double disc type, and so on. Guo Hui et al. [1] designed a conical wheel-type earth-covering suppresser that integrates earth-covering and suppression functions. At the same time, the

test was carried out with the thickness of the covering soil and the compactness of the soil as the test indexes, and the test results were analyzed to determine the optimal structural parameters of the conical wheel covering soil suppresser so as to simplify the structure of the machine and improve the stability of the covering soil and suppress operation. Du Wenbin et al. [2] designed a cigar tobacco leaf adjustable seedling bed ridging and film laying machine, which integrates soil covering operation with rotary tillage, film laying, and other operation machinery. According to the operation requirements of the operation process, the structure of the soil-covered disc and the key structural parameters were designed. Guo Zhenhua et al. [3] designed an orchard rear suspension ditching and fertilizing soil covering machine and carried out integrated design research on ditching, fertilizing, and soil covering devices. By analyzing the process of soil covering operation and the effect of soil cutting and turning, a double-disc soil covering mechanism with an adjustable dip angle in the forward direction and vertical direction of the soil covering disc is designed. Geng Yuanle et al. [4] designed a chisel-type ditching and covering device with a soil sealing function for maize no-tillage planter to solve the problems of poor soil fluidity and poor soil restoration effect in the process of maize no-tillage sowing. The theoretical analysis and design of the overburden plate are carried out, and its key parameters are determined. Ke Li et al. [5] designed a soil-covering device for sugarcane horizontal planting to solve the problem of the sugarcane soil-covering device not meeting the agronomic standards of sugarcane intercropping. Through the theoretical analysis of the covering soil device, the quadratic orthogonal rotation test, and the regression analysis of the results, the key factors affecting the thickness of the covering soil were determined. Deng Xing [6] designed a sand soil continuous trenching and covering machine suitable for desertification soil in southern Xinjiang. He mainly completed the force analysis of the trenching and covering device, as well as the influence of the structural and working parameters of the trenching device on the trenching resistance. Based on the discrete element method simulation of the trenching and covering device's operation process, the optimal parameters of the trenching device were determined, but no research was conducted on the relevant parameters of the trenching and covering device. Finally, the parameters of the covering soil device were optimized. Based on the above literature and other references, it can be seen that [7–17], now, most soil covering devices are applied to crop cultivation (such as corn and cotton), and there is relatively little research on the design of soil covering devices suitable for tree planting machines. At the same time, in the research of soil covering devices, most of the current research on soil covering devices integrates soil covering with compaction, trenching, and other devices. The research on soil covering devices alone is not deep enough, and most of them still design and analyze the basic structure of soil covering devices, lacking optimization design of soil covering devices.

In order to solve the problem of low survival rate and low survival rate of seedlings caused by low soil backfilling rate of tree planting machines, and to meet the operation requirements of different kinds of seedlings in different environments, this paper designs a symmetrical double disc type operation width, forward angle and soil depth adjustable tree planting machine covering device, and then analyzes and studies its covering process. The range of key structural parameters was determined. At the same time, the simulation test was designed, and the results of the test were analyzed, and the key structural parameters of the covering device were optimized.

## 2. The Overall Structure and Working Principle of Tree Planting Machine Soil Covering Device

### 2.1. Overall Structure of Tree Planting Machine Soil Covering Device

In view of the large area of desertification in southern Xinjiang, the demand for continuous and efficient tree planting machines, and the characteristics of easy return of sand and soil, the existing tree planting machine ditching method is the secondary ditching operation mode, that is, the two continuous ditching makes the tree planting ditch section

rectangular. In order to meet the soil covering requirements of the existing rectangular tree planting ditch, the soil covering device of the tree planting machine with adjustable soil depth, forward angle, and working width is designed as shown in Figure 1. The soil covering device of the tree planting machine is mainly composed of an installation frame, height adjustment plate, angle adjustment shaft, width adjustment shaft, soil covering plate, and other structures. The installation frame is fixed to the tree-planting machine frame. The lower side of the installation frame is connected with the height adjustment plate by a spring and long-axis bolt, and the height adjustment can be realized by screwing the nut on the long-axis bolt. The height adjustment plate is channel steel with a 'T' shape structure. The width adjustment shaft is installed horizontally on the lower side of the height adjustment plate. The nut on the width adjustment shaft can realize the adjustment of the width of the covering soil. The width adjustment shaft is connected with the angle adjustment shaft through the fastening bolt, and the covering plate is installed on the lower side of the angle adjustment shaft. The bolt on the shaft can be adjusted by loosening the width to adjust the angle to adjust the different steering angles of the shaft to control the forward angle of the covering plate.

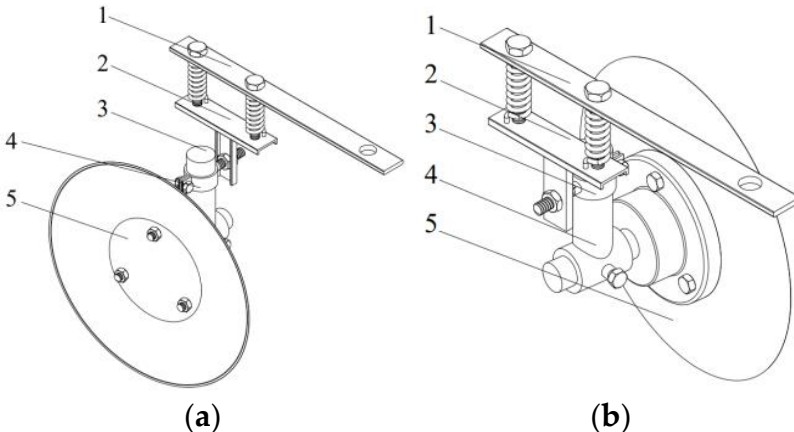

(**a**)          (**b**)

**Figure 1.** Structure diagram of soil covering device of tree planting machine function: 1—Installation frame, 2—Height adjustment plate, 3—Angle adjustment shaft, 4—Width adjustment shaft, 5—Covering plate. (**a**) Tree planting machine left soil covering device; (**b**) Tree planting machine right soil covering device.

*2.2. Working Principle of Tree Planting Machine Soil Covering Device*

The operation process of the tree planting machine's soil covering device is shown in Figure 2. The forward direction of the tree planting machine is in the X direction shown in the diagram. The soil covering device is installed on the tree planting machine frame at the left and right rear ends, and the soil covering discs on the left and right soil covering devices are installed at a 'V'-shaped angle. The soil covering device starts the soil covering operation after the tree planting machine completes the rectangular trenching operation (secondary continuous trenching), seedling planting operation, and other operations. During the soil covering operation, the left and right soil trays will dig trenches and plow the soil, squeezing it into the tree planting ditch to complete the soil covering operation for tree seedlings. Ultimately, it achieves the function of covering soil.

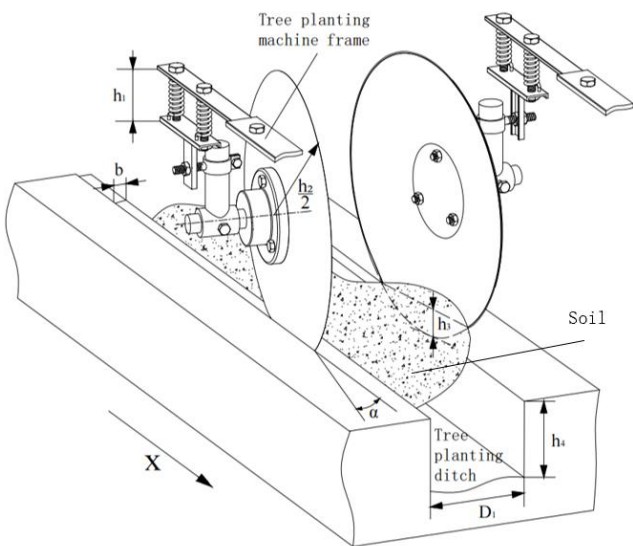

**Figure 2.** Schematic diagram of the operation process of the soil covering device. Note: $h_1$ is the height difference between the height adjustment plate and the mounting frame, mm; $h_2$ is the diameter of the overburden plate, mm; $h_3$ is the buried depth of the soil-covered disc, mm; $h_4$ is the ditching depth, mm; $D_1$ is the ditching width of the tree planting machine, mm; $\alpha$ is the advancing angle of the covering plate, °; b is the working width, mm.

The soil backfilling rate and depth of the soil covering device of the tree planting machine during the soil covering operation will directly affect the lodging rate and survival rate of the tree planting machine. Through the analysis of the above homework process, it can be concluded that the structural factors that have the greatest impact on soil backfilling rate and depth in the covering device are the covering width (straight-line distance between the left and right covering discs) and the forward angle of the covering disc $\alpha$. The depth of the soil tray entering the soil is h3. Therefore, in order to ensure that the soil backfilling rate and depth of the covering device meet the operational requirements, thereby improving the survival rate of seedlings and reducing the lodging rate, this article needs to address the key structural parameters of the covering device, such as the covering width $D_2$ and the forward angle $\alpha$. Optimize the design based on the depth of excavation, such as $h_3$.

### 3. Design of Key Structural Parameters of Soil Covering Device

*3.1. Soil Covering Width of Soil Covering Device*

The width of the soil cover device refers to the straight-line distance between the central axis of the soil cover plates on both sides of the tree-planting machine. The schematic diagram of the soil cover device from the top is shown in Figure 3. The analysis of the soil covering operation shows that the soil covering width $D_2$ of the soil covering device is mainly determined by the ditching width $D_1$ of the tree planting machine. When the covering width $D_2$ is less than the ditching width $D_1$, the covering plate will be suspended above the planting ditch, and the covering operation can not be completed. When the width of covering soil $D_2$ is much larger than the width of ditching $D_1$, that is, the width of operation b is too large, the amount of covering soil that the covering wheel turns into the planting ditch can not meet the requirements of seedling planting. The analysis of the amount of covering soil of the covering wheel shows that when the width of operation b is 80–100 mm, it can better meet the requirements of covering soil.

In order to meet the operation requirements so that the soil covering plate cannot be suspended above the tree planting ditch, and to meet the requirements that the ditching width $D_1$ is 400 mm and the operation width b is 80–100 mm, the soil covering width $D_2$ of the soil covering device is designed to be 560–600 mm.

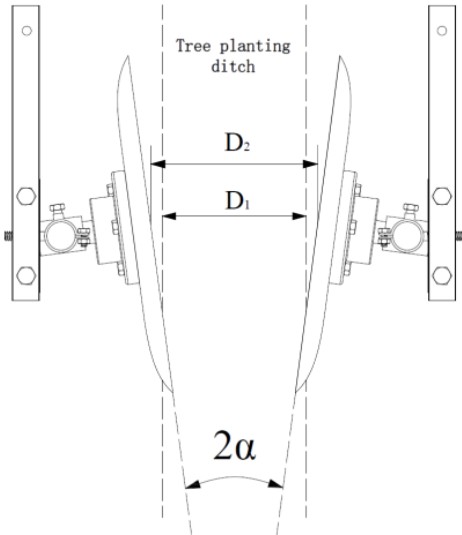

**Figure 3.** Overlooking schematic diagram of soil covering device. Note: $D_2$ is the width of the tree planting machine, mm.

### 3.2. The Forward Angle of the Overburden Plate

The forward angle $\alpha$ of the tree-planting machine's covering plate refers to the angle between the forward direction of the tree-planting machine and the covering plate. The larger the forward angle $\alpha$ of the covering plate is, the stronger the soil-turning ability is, but the greater the traction resistance is. According to the requirements of the amount of soil, the amount of soil, the diameter of the soil wheel, and the width of the soil, the forward angle $\alpha$ of the soil plate is designed to be 40~50°.

### 3.3. The Buried Depth of Soil Covering Device

The burial depth of the soil cover device refers to the depth at which the soil cover plate rotates and cuts into the plowed soil. The burial depth can be expressed as the amount of compression of the soil cover plate on the plowed soil, which indirectly represents the size of the soil backfill rate. The front view of the soil covering device is shown in Figure 4.

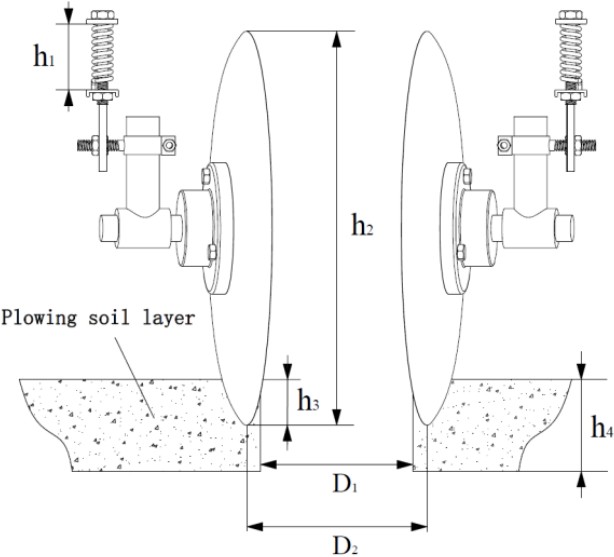

**Figure 4.** Forward view of soil covering device.

According to the planting machine ditching depth $h_4$ is 200–300 mm, ditching width $D_1$ is 400 mm, operation width b is 80–100 mm operation requirements. It is difficult to meet the installation requirements when the diameter $h_2$ of the overburden plate is too large; when the diameter $h_2$ of the soil covering plate is too small, the operation experience that the cutting depth of the soil covering plate to the plowing soil is not enough leads to less soil backfill, and the diameter $h_2$ of the soil covering plate is determined to be 300 mm. Combined with the geometric relationship of each parameter in the operation process of the soil covering device shown in Figure 2, the buried depth $h_3$ is shown in Formula (1).

$$h_3 = \frac{h_2}{2} - \sqrt{\left(\frac{h_2}{2}\right)^2 - \left(\frac{b}{2\sin\alpha}\right)^2} \tag{1}$$

Among them, the diameter of the covering plate $h_2$ is 300 mm, the working width b is 80–100 mm, and the advancing angle of the covering plate $\alpha$ is 40–50°. Therefore, the depth of the soil $h_3$ is 10–20 mm from the above (1).

The analysis of the influencing factors of the buried depth $h_3$ shows that under the condition that the total mass of the tree planting machine is fixed and the installation frame of the soil covering device is firmly fixed on the frame of the tree planting machine, the buried depth $h_3$ is mainly determined by the height difference $h_1$ between the height adjustment plate and the installation frame and the diameter $h_2$ of the soil covering plate.

When the diameter of the covering plate $h_2$ is a fixed value of 300 mm, the adjustment of the depth $h_3$ is mainly achieved by adjusting the height difference $h_1$ by screwing the nut. Therefore, in the case of a certain diameter $h_2$ of the covering plate and according to the operation requirements of the depth $h_3$ adjustment range of 10–20 mm, the height difference $h_1$ adjustment range can be calculated and designed to be 10–20 mm.

## 4. Simulation Test of Soil Covering Process

In recent years, the Discrete Element Method (EDEM) has gradually begun to be widely applied in the field of agricultural machinery research that interacts with soil [18,19]. The discrete element method can simulate the interaction and motion process of various elements [20], and when used to study the interaction between soil and soil touching components, the simulation results are highly consistent with the actual situation [21,22]. This study establishes an EDEM simulation model to explore the influence of structural parameters of the double disc soil covering device of a tree planting machine on the soil backfill rate.

### 4.1. Simulation Model Establishment

4.1.1. Establishment of Particle Model and Geometric Model

In the process of establishing particle and geometric models, reference was made to relevant literature on EDEM simulation research of tree planting machines. Solidworks 2021 software was used to establish a three-dimensional model of tree planting ditches, which was saved in .igs format and imported into EDEM 2020 software. The total length of the soil model was 5000 mm, the width was 1400 mm, and the height was 500 mm. The rectangular tree planting ditch is 5000 mm long, 400 mm wide, and 250 mm high. Set the average radius of sand particles to 10 mm and use a nonsliding Hertz Mindlin contact model between particles and the soil wheel model, as well as between particles. The number of soil particles is set to be unlimited, and the generation speed is 1000 per second until the filling of the whole soil model is completed. Finally, the tree planting ditch soil model shown in Figure 5 is generated in EDEM. At the same time, Solidworks software is used to establish the three-dimensional modeling of the soil covering device, which is saved as .igs format and imported into EDEM software. In order to shorten the simulation calculation time, on the premise of not affecting the calculation results, the auxiliary parts that do not participate in the soil covering of the tree planting ditch are deleted, and only

the left and right soil covering plates are retained for simulation calculation, as shown in Figure 6.

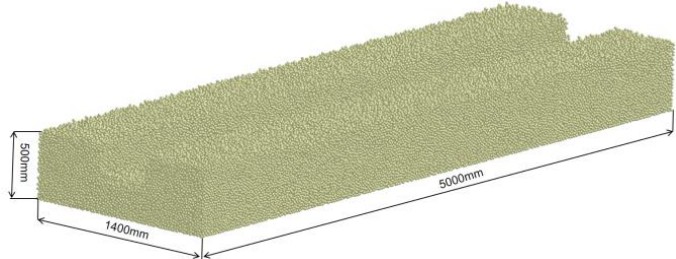

**Figure 5.** Tree-planting ditch soil model.

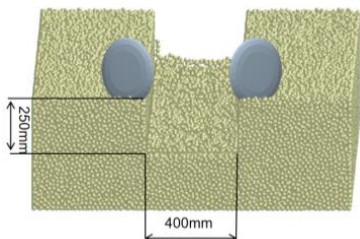

**Figure 6.** Interaction model of soil covering plate and planting ditch.

4.1.2. Simulation Parameter Setting

To ensure the accuracy of the simulation results of the soil covering device, it is necessary to set the parameters of the soil, the material of the soil covering device itself, and the parameters of mutual contact. Reference relevant literature [23–25] on the parameters of the material itself, such as density, Poisson's ratio, and shear modulus. The contact parameters such as collision recovery coefficient, static friction coefficient, and rolling friction coefficient are set in Table 1.

**Table 1.** Discrete element simulation parameters.

| Type | Parameters | Numerical Value |
|---|---|---|
| Soil particle | Poisson ratio | 0.3 |
| | Density (kg/m$^3$) | 1616 |
| | Shear modulus (Pa) | $5 \times 10^7$ |
| Soil covering device (65Mn) | Poisson ratio | 0.3 |
| | Density (kg/m$^3$) | 7861 |
| | Shear modulus (Pa) | $7 \times 10^{10}$ |
| Sand-sand | Restitution coefficient | 0.279 |
| | Coefficient of static friction | 0.723 |
| | Coefficient of rolling friction | 0.393 |
| Sand—65Mn | Restitution coefficient | 0.101 |
| | Coefficient of static friction | 0.847 |
| | Coefficient of rolling friction | 0.153 |
| Other parameters | Acceleration of gravity (m/s$^2$) | 9.8 |
| | Forward speed (m/s) | 1.5 |

### 4.2. Experimental Design

Test Factors and Indicators

For the soil covering device of a tree-planting machine, the width of the soil cover, the angle of advancement, and the depth of penetration are the direct factors that affect its soil covering performance. These parameters have a direct impact on the lodging rate of tree

seedlings in the tree planting machine. Therefore, the width of the soil cover, the angle of advancement, and the depth of penetration were selected as experimental factors for the experiment, and theoretical analysis and preliminary testing were conducted using the Box Behnken test method for a three-factor and three-level simulation experiment [26]. Based on the parameter range of the previously designed soil cover device, the factor code table is shown in Table 2.

**Table 2.** Test factor coding.

| Coding | Factor | | |
| --- | --- | --- | --- |
| | Overburden Width $D_2$ (mm) | Forward Angle $\alpha$ (°) | Operating Depth $h_3$ (mm) |
| −1 | 560 | 40 | 10 |
| 0 | 580 | 45 | 15 |
| 1 | 600 | 50 | 20 |

The soil backfill rate y1 was selected as the test index to carry out the simulation test. The soil backfill rate y1 refers to the ratio of the soil excavated from the tree planting ditch excavated by the tree planting machine to the soil backfilled to the tree planting ditch after the soil covering device of the tree planting machine. In this paper, the soil backfill rate y1 is calculated by the ratio of the cross-sectional area $S_1$ of the tree planting ditch and the cross-sectional area $S_2$ of the soil covering device after the stable operation of the soil covering device. Because the cross-section of the tree planting ditch is the same as the bottom of the cross-section of the soil covering, the solid soil backfill rate is the ratio of the height of the tree planting ditch to the height of the soil covering. The formula is as follows:

$$y1 = \frac{S_2}{S_1} \times 100\% \tag{2}$$

$$y1 = \frac{D_1 H_2}{D_1 H_1} \times 100\% \tag{3}$$

$$y1 = \frac{H_2}{H_1} \times 100\% \tag{4}$$

Note: $H_1$ is the depth of planting ditch; $H_2$ is the depth of covering soil.

*4.3. Experiment Results and Analysis*

Different covering widths, advancing angles, and buried depths are selected to simulate the covering operation. The forward speed of the covering plate is set to 1.5 m/s, and each group of tests advances 5 m, as shown in Figure 7a, which is the covering process of the covering plate. After the simulation, the height of the soil covering the tree planting ditch was measured using the ruler tool of EDEM. In the stable soil covering stage of 3 m–4 m, a point was selected every 0.3 m, and a total of 3 points were selected, as shown in the triangle mark in the figure. The average soil covering height of these three points was taken as an experimental result, as shown in Figure 7b.

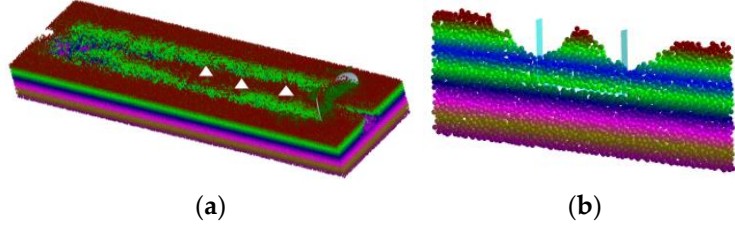

(**a**)                         (**b**)

**Figure 7.** Simulation results and analysis of soil covering device. Note:The white triangle represents the sampling points; The blue box represents the tree planting ditch. (**a**) Soil covering process; (**b**) Analysis of results.

A total of 17 groups of tests were carried out, each test was repeated three times, and the test results were taken as the average value. The test scheme and results are shown in Table 3. In Table 3, $X_1$, $X_2$ and $X_3$ were the coded values of overburden width, advancing angle and buried depth, respectively.

**Table 3.** Test scheme and results.

| Test Serial Number | Factor | | | Soil Backfill Rate y1 |
|:---:|:---:|:---:|:---:|:---:|
| | $X_1$ | $X_2$ | $X_3$ | |
| 1 | −1 | −1 | 0 | 62.7 |
| 2 | 1 | −1 | 0 | 68.1 |
| 3 | −1 | 1 | 0 | 69.7 |
| 4 | 1 | 1 | 0 | 70.6 |
| 5 | −1 | 0 | −1 | 80.3 |
| 6 | 1 | 0 | −1 | 81.2 |
| 7 | −1 | 0 | 1 | 83.5 |
| 8 | 1 | 0 | 1 | 84.7 |
| 9 | 0 | −1 | −1 | 77.8 |
| 10 | 0 | 1 | −1 | 86.9 |
| 11 | 0 | −1 | 1 | 86.3 |
| 12 | 0 | 1 | 1 | 88.7 |
| 13 | 0 | 0 | 0 | 84.2 |
| 14 | 0 | 0 | 0 | 82.7 |
| 15 | 0 | 0 | 0 | 85.1 |
| 16 | 0 | 0 | 0 | 83.5 |
| 17 | 0 | 0 | 0 | 84.5 |

*4.4. Analysis of Experimental Results*

Analysis of variance and regression equation: The simulation results were substituted into Design-Expert for regression analysis and analysis of variance. The results showed that when the quadratic regression was used, the model was significant and the lack of fit was not significant ($p < 0.05$), and the model was established. From the analysis of variance in Table 4, the coefficient of determination $R^2 = 0.9916$, the coefficient of variation CV = 1.32%, and the corrected coefficient of determination $R^2$adj = 0.9808 were obtained, indicating that the test had good reliability and the fitting equation had high reliability. Among them, $X_1$, $X_2$, $X_3$, $X_2X_3$, $X_1{}^2$, $X_2{}^2$, $X_3{}^2$ have a significant effect on the index Y, and the regression equation is as follows:

$$y1 = 84 + 1.05X_1 + 2.63X_2 + 2.13X_3 - 1.13X_1X_2 + 0.08X_1X_3 - 1.68X_2X_3 - 9.36X_1^2 - 6.86X_2^2 + 7.79X_3^2 \tag{5}$$

**Table 4.** Variance analysis table.

| Source | Sum of Squares | Df | Mean Square | F-Value | *p*-Value |
|:---:|:---:|:---:|:---:|:---:|:---:|
| Model | 914.71 | 9 | 101.63 | 91.62 | <0.0001 |
| $X_1$ | 8.82 | 1 | 8.82 | 7.95 | 0.0258 |
| $X_2$ | 55.12 | 1 | 55.12 | 49.69 | 0.0002 |
| $X_3$ | 36.12 | 1 | 36.12 | 32.56 | 0.0007 |
| $X_1X_2$ | 5.06 | 1 | 5.06 | 4.56 | 0.0700 |
| $X_1X_3$ | 0.02 | 1 | 0.02 | 0.02 | 0.8908 |
| $X_2X_3$ | 11.22 | 1 | 11.22 | 10.11 | 0.0155 |
| $X_1{}^2$ | 369.07 | 1 | 369.07 | 332.71 | <0.0001 |
| $X_2{}^2$ | 198.29 | 1 | 198.29 | 178.75 | <0.0001 |
| $X_3{}^2$ | 255.34 | 1 | 255.34 | 230.19 | <0.0001 |
| Residual | 7.765 | 7 | 1.11 | | |
| Lack of Fit | 4.325 | 3 | 1.44 | 1.68 | 0.3081 |
| Pure Error | 3.44 | 4 | 0.86 | | |
| Cor Total | 922.48 | 16 | | | |

According to the analysis of variance of the regression model, it can be seen that the three factors selected have a very significant impact on the soil backfill rate ($p < 0.05$). The order of influence is $X_3$ (forward angle) > $X_2$ (operating depth) > $X_1$ (overburden width), and the interaction between $X_2$ and $X_3$ is also significant.

*4.5. Optimization Choice for Parameters*

Using the optimization module in Design-Expert 13 software, the measured value of soil backfill rate is 89% as the target value, and a set of solutions similar to the measured value is obtained: covering width $X_1 = 600$ mm, advancing angle $X_2 = 45°$, buried depth $X_3 = 20$ mm. In order to verify the reliability of the set of parameters, three sets of simulation tests were carried out with the above parameters as EDEM simulation parameters. The average value of the soil backfill rate was 88.6%, and the relative error with the measured value was 0.40%, indicating that the set of simulation parameters had high reliability and authenticity.

## 5. Prototype Test Verification

After completing the 3D modeling of the soil cover plate of the tree planting machine and optimizing the parameters of the soil cover plate, the prototype was developed at Tumushuke Yinfeng Modern Agricultural Equipment Co., Ltd. which located in Tumushuk City, Xinjiang, China. The field experiment chose to plant trees in the desertified soil of Tumushuke in southern Xinjiang in the autumn. Different working parameters were selected, and the accuracy of the discrete element simulation was verified by comparing the simulation results with the field experiment results.

As shown in Figure 8, a field experiment was carried out in Tumushuke City, Xinjiang Production and Construction Corps. According to the 120 kW matching power selected by the tree planting machine, a 1204 tractor is selected for tree planting operation. The operation speed is controlled by the tractor to be 1.5 m/s, and a tree-planting ditch with a depth of 250 mm and a width of 400 mm is cultivated. According to the simulation results, the center spacing of the overburden plate is set by adjusting the width adjustment shaft, that is, the width of the overburden plate is 600 mm, the forward angle of the overburden plate is set to 45° by adjusting the angle adjustment shaft, and the depth of the overburden plate is set to 20 mm by adjusting the width adjustment shaft. When the tree planting machine runs stably, the measured soil backfill rate is 88.4%. The error between the experimental results and the simulation results is small, which proves that the tree planting machine has high reliability, stable operation, and meets the design requirements.

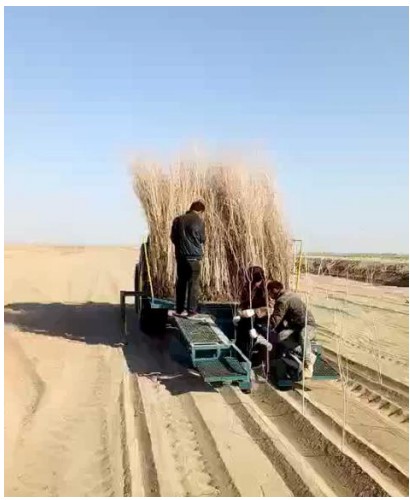

**Figure 8.** Field experiment of prototype.

## 6. Conclusions

(1) According to the soil covering requirements of the existing rectangular tree planting ditch, the soil covering device of the tree planting machine with adjustable soil depth, forward angle, and operation width is designed. The height adjustment can be realized by screwing the nut on the long-axis bolt. The width adjustment shaft installed laterally at the lower side of the height adjustment plate can adjust the width of the covering soil. The width adjustment shaft is connected with the angle adjustment shaft by fastening bolts. The bolts on the width adjustment shaft are loosened to adjust the different steering angles of the angle adjustment shaft to control the forward angle of the covering soil plate.

(2) By using the simulation software EDEM, the simulation model of the interaction between soil and soil covering device is established. Taking the soil coverage rate as the index, the influence of the structural parameters of the double disc soil covering device of the tree planting machine on the soil backfill rate and soil coverage rate is discussed. Through experimental analysis, the optimum soil covering width is 600 mm, the forward angle is 45°, and the soil depth is 20 mm. At this time, the soil backfill rate of the tree planting ditch can reach 88.4%.

(3) The research in this paper also has certain limitations. This paper mainly studies the use of the soil covering device of the tree planting machine in the sandy loam environment. The next step can be carried out to study whether it is suitable for other types of soil, as well as in different weather conditions, soil moisture, and other conditions. This paper further improves the applicability of the device.

**Author Contributions:** Conceptualization, Z.J. and L.Z.; methodology, X.H.; experiment, X.W. and W.L.; data curation, Z.J.; writing—original draft preparation, X.W. and Z.J.; writing—review and editing, L.Z. and Z.J. All authors have read and agreed to the published version of the manuscript.

**Funding:** This work has been supported by the financial science and technology plan project of Xinjiang Production and Construction Corps (Grant No. 2022CB006-01).

**Institutional Review Board Statement:** Not applicable.

**Informed Consent Statement:** Not applicable.

**Data Availability Statement:** Data is contained within the article.

**Acknowledgments:** First of all, we would like to thank the mentor for their guidance and technical support. We are also very grateful to the brothers and sisters who contributed to this article. Finally, we thank editors and anonymous reviewers for providing useful suggestions for improving the quality of this article.

**Conflicts of Interest:** Author Wenchun Li was employed by the company Tumushuke Yinfeng Modern Agricultural Equipment Co., Ltd. The remaining authors declare that the research was conducted in the absence of any commercial or financial relationships that could be construed as a potential conflict of interest.

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
