# Peer review of "Optimization Design and Experimentation of a Soil Covering Device for a Tree Planting Machine"

_agriculture, doi:10.3390/agriculture14030346_

Round 1
Reviewer 1 Report
Comments and Suggestions for Authors
1. How does varying the depth of soil coverage affect tree growth and survival rates
2. How does the soil covering influence water retention in the planting area
3. What is the optimal balance between water retention and drainage for tree growth
4. How does the design influence the uniformity of soil coverage around each tree
5. How does the performance of the soil covering device vary across different soil types
6. Can the device be adapted for use in a variety of environmental conditions
7. What is the economic feasibility of implementing the optimized soil covering device on a large scale
8. How does the device impact the overall cost and benefits of tree planting operations
9. What are the results of field testing the soil covering device in various locations and climates
Author Response
Dear reviewers
Re: Manuscript lD: agriculture-2805075 and Title: Optimization Design and Experimentation of Soil Covering Device for Tree Planting Machine
Thank you for your letter and the reviewers'comments concerning our manuscriptentitled “Optimization Design and Experimentation of Soil Covering Device for Tree Planting Machine” (agriculture-2805075). Those comments are valuable and very helpful. We haveread through comments carefully and have made corrections. Based on theinstructions provided in your letter, we uploaded the file of the revised manuscript.Revisions in the text are shown using blue highlight for additions, and strikethroughfont for deletions. The responses to the reviewer's comments are marked in red andpresented following.
We would love to thank you for allowing us to resubmit a revised copy of themanuscript and we highly appreciate your time and consideration.
Sincerely.
Xun Wu
- How does varying the depth of soil coverage affect tree growth and survival rates
Response: The buried depth of seedlings is directly related to the respiration and stability of roots. If the soil is buried too deep, the root system is difficult to get full breathing, thus affecting the survival rate ; if the buried soil is too shallow, the roots will be exposed, which will easily lead to lodging and not easy to survive. Therefore, when planting seedlings, it is necessary to use hands to straighten up, let the roots stretch, and maintain an appropriate burial depth, generally no more than 10 centimeters, and then water the soil to maintain an appropriate degree of moisture.
- How does the soil covering influence water retention in the planting area
Response: Generally, the influencing factors of water retention performance in planting areas are mainly related to soil covering materials. Choosing loose and breathable materials, such as straw, wheat straw, sawdust, etc., can effectively maintain soil ventilation and avoid the growth of bacteria in the soil. At the same time, these materials also have good water retention, which can slow down the evaporation rate of the soil and keep the soil moist. This paper does not carry out research in this area, and subsequent research can be carried out on this issue.
- What is the optimal balance between water retention and drainage for tree growth
Response: The key to the best water retention and drainage balance of tree growth is that the appropriate soil conditions and appropriate water management, loose soil in sandy loam, not easy to accumulate water, and need a lot of water to keep the soil wet, and the soil water content of 60 % to 80 % of the maximum water holding capacity is the most appropriate.
- How does the design influence the uniformity of soil coverage around each tree
Response: When the soil covering device is working, the soil is lifted and thrown at the same time, forming a soil block. The soil block will fall on the seed or seedling of the crop with the movement of the soil covering device to achieve the soil covering effect. The thickness and uniformity of the covering soil depend on the setting of the adjusting device, including the depth of the covering soil and the adjustment of the forward angle.
- How does the performance of the soil covering device vary across different soil types
Response: We are very sorry that the soil covering device we designed is currently only for sandy loam soil, and other types of soil have not been studied.
- Can the device be adapted for use in a variety of environmental conditions
Response: At present, the device is only suitable for sandy loam, only tested in sandy loam environment, and no related tests are carried out for other environmental conditions.
- What is the economic feasibility of implementing the optimized soil covering device on a large scale
Response: At present, the device is still in a small-scale trial, compared with the traditional tree planting, it effectively realizes the saving of planting cost.
- How does the device impact the overall cost and benefits of tree planting operations
Response: The soil covering plate of the tree planting machine designed in this paper improves the speed and efficiency of tree planting, reduces the labor cost and increases the survival rate of trees. So as to reduce the cost of tree planting and improve the income of tree planting operations.
- What are the results of field testing the soil covering device in various locations and climates
Response: We did not test under different environmental and climatic conditions, only under a single condition. The test results show that the soil covering results can meet the needs of tree planting.

Reviewer 2 Report
Comments and Suggestions for Authors
Review
The manuscript appears to be clear, relevant for the field, and presented in a well-structured manner. The paper is also well-structured, and appears to be a well-written and informative contribution to the field of agricultural engineeringI report below the pros and cons of the article, which, in my opinion, still needs to be reconsidered after major revisions.
Aim of the paper, its main contributions and strengths
The aim of the paper is to address the challenges of low survival rates and lodging of seedlings caused by inadequate soil backfilling rates in tree planting operations, particularly in desert areas of southern Xinjiang. The main contributions and strengths of the paper can be outlined as follows:
Innovative Design Approach: The paper introduces an innovative soil covering device for tree planting machines with adjustable soil covering width, forward angle, and soil depth. This design addresses the specific requirements of tree planting in desert areas, aiming to improve the soil covering effect and reduce the lodging rate of seedlings.
Advanced Simulation and Optimization: The study leverages discrete element simulation software (EDEM) to conduct an orthogonal test, using the width of the covering soil and the inclination angle as test factors and the soil backfill rate as the response index. This approach demonstrates a commitment to advanced engineering and simulation techniques for optimizing agricultural machinery.
Optimal Parameter Determination: Through regression analysis of the test results, the paper determines the optimal parameter combination for the covering soil device, including the width of the covering soil, forward angle, and soil depth. This empirical optimization contributes to practical and actionable insights for improving tree planting efficiency.
The findings of the study hold practical significance for the agricultural industry, particularly for manufacturers of tree planting machines and equipment. The optimized soil covering device has the potential to enhance the soil covering rate of tree planting machines and reduce the lodging rate of seedlings, addressing critical challenges in tree planting operations. By focusing on improving tree planting efficiency in desert areas and addressing land desertification, the paper aligns with the broader goals of environmental conservation and sustainable land management, making it relevant to agricultural and environmental stakeholders. Overall, I believe that the paper's innovative design approach, advanced simulation techniques, empirical optimization, and practical implications for agricultural equipment make it a valuable contribution to the field of agricultural machinery and tree planting operations.
Weakness and aspects that could benefit from further improvement
While the paper presents valuable contributions, it's important to consider potential areas of weakness and aspects that could benefit from further scrutiny:
1. Testability of Hypothesis: The paper lacks a clearly stated hypothesis or research question, which is essential for guiding the study and evaluating its outcomes. A well-defined hypothesis would provide a clear focus for the research and enable the assessment of its validity.
2. Methodological Inaccuracies: The paper could benefit from a more detailed description of the methodology, particularly regarding the experimental design and simulation process. Providing a comprehensive overview of the experimental setup, data collection methods, and simulation parameters would enhance the transparency and reproducibility of the study.
3. Missing Controls: The absence of a discussion on potential confounding variables or controls in the experimental design is a notable weakness. Addressing the potential influence of external factors on the soil covering process and backfill rate would strengthen the robustness of the findings.
4. Validation of simulation results: Although the use of the EDEM simulation model is a strength, the paper could elaborate on the validation of simulation results against real data or empirical observations. Ensuring the accuracy and reliability of the simulation model is critical to drawing meaningful conclusions. Results on this aspect are lacking as well as discussions with what exists in the literature. These aspects, i.e., test verification of the prototype, which is completely to be developed, and discussions with what exists in the literature, need to be implemented. As it is now, I do not think it is sufficient. The paper could benefit from a comparative analysis of the proposed soil covering device with existing or traditional methods. A comparative assessment would provide valuable insights into the potential advantages and limitations of the new device in relation to established practices.
5. The conclusion is concise but could benefit from a brief restatement of the main findings and their significance. Verify that the conclusion includes the following:
· Restate your hypothesis or research question;
· Restate your main findings;
· Tell the reader how your study contributes to the existing literature;
· Identify any limitations of your study;
· State future research directions/recommendations;
Author Response
Dear reviewers
Re: Manuscript lD: agriculture-2805075 and Title: Optimization Design and Experimentation of Soil Covering Device for Tree Planting Machine
Thank you for your letter and the reviewers'comments concerning our manuscriptentitled “Optimization Design and Experimentation of Soil Covering Device for Tree Planting Machine” (agriculture-2805075). Those comments are valuable and very helpful. We haveread through comments carefully and have made corrections. Based on theinstructions provided in your letter, we uploaded the file of the revised manuscript.Revisions in the text are shown using blue highlight for additions, and strikethroughfont for deletions. The responses to the reviewer's comments are marked in red andpresented following.
We would love to thank you for allowing us to resubmit a revised copy of themanuscript and we highly appreciate your time and consideration.
Sincerely.
Xun Wu
- Testability of Hypothesis: The paper lacks a clearly stated hypothesis or research question, which is essential for guiding the study and evaluating its outcomes. A well-defined hypothesis would provide a clear focus for the research and enable the assessment of its validity.
Response:Our research questions are as follows : In order to solve the problem of low survival rate of seedlings caused by low soil backfilling rate of tree planting machine, and to meet the operation requirements of different types of seedlings in different environments, a symmetrical double-disc soil covering device with adjustable working width, forward angle and buried depth of tree planting machine is designed.
- Methodological Inaccuracies: The paper could benefit from a more detailed description of the methodology, particularly regarding the experimental design and simulation process. Providing a comprehensive overview of the experimental setup, data collection methods, and simulation parameters would enhance the transparency and reproducibility of the study.
Response:To describe the experimental method in detail, see the latest manuscript.
- Missing Controls: The absence of a discussion on potential confounding variables or controls in the experimental design is a notable weakness. Addressing the potential influence of external factors on the soil covering process and backfill rate would strengthen the robustness of the findings.
Response:At present, the working environment of the design of the device is relatively single, only for sand, so the influence of relevant environmental factors on the device is not considered.
- Validation of simulation results: Although the use of the EDEM simulation model is a strength, the paper could elaborate on the validation of simulation results against real data or empirical observations. Ensuring the accuracy and reliability of the simulation model is critical to drawing meaningful conclusions. Results on this aspect are lacking as well as discussions with what exists in the literature. These aspects, i.e., test verification of the prototype, which is completely to be developed, and discussions with what exists in the literature, need to be implemented. As it is now, I do not think it is sufficient. The paper could benefit from a comparative analysis of the proposed soil covering device with existing or traditional methods. A comparative assessment would provide valuable insights into the potential advantages and limitations of the new device in relation to established practices.
Response:We have read the relevant literature. At present, most of the soil covering devices are used in crop planting ( corn, cotton, etc. ), and there are relatively few studies on the design of soil covering devices suitable for tree planting machines. At the same time, in the research of the soil covering device, most of the current research on the soil covering device will consider the integration of the soil covering device, the pressure device, the ditching device and so on. The research on the soil covering device alone is not deep enough, and most of them still design and analyze the basic structure of the soil covering device and lack the optimization design of the soil covering device.
- The conclusion is concise but could benefit from a brief restatement of the main findings and their significance. Verify that the conclusion includes the following:
- Restate your hypothesis or research question;
- Restate your main findings;
- Tell the reader how your study contributes to the existing literature;
- Identify any limitations of your study;
- State future research directions/recommendations;
Response:The conclusion has been supplemented in the latest paper.

Reviewer 3 Report
Comments and Suggestions for Authors
Line 38: “In order to solve the problem of low survival rate and lodging rate...”. Shall it be “In order to solve the problem of low survival rate and high lodging rate...”?
Line 115: “figure 2” or “Figure. 2” or “Figure 2”? Please be consistent through out the manuscript.
Line 118, 120: “' V ' -shaped angle”, “ ( secondary continuous ditching )” etc. Please check the usage of space through out the manuscript and make sure it is appropriate.
Line 136: “width of the soil cover”. May add “width of the soil cover D_2”.
Line 185: “h3”, use subscript. Please check all the subscripts in the manuscript.
Figure 5: May be better to mark the dimensions.
Table 2: Make sure the table is on the same page.
Line 258: “edem” to “EDEM”. Check the use of capitalization in the manuscript.
Line 259: “and a total of 3 points were selected”. May be better to mark the points in Figure 7.
Line 261: “Figure 2.” should be “Figure 7.” Please be consistent throughout the manuscript.
Line 306: “1.5m / s” to “1.5 m/s” Please check the usage of space throughout the manuscript and make sure it is appropriate.
Line 310: “45 °” to “45°” Please check the usage of space throughout the manuscript and make sure it is appropriate.
Line 333-335, 341: Check the usage of space.
General comments: In the simulation, the tree seedlings were not included. Do you expect the results to differ significantly with and without the presence of the tree seedlings? If so, why?
Comments on the Quality of English LanguageNA
Author Response
Dear reviewers
Re: Manuscript lD: agriculture-2805075 and Title: Optimization Design and Experimentation of Soil Covering Device for Tree Planting Machine
Thank you for your letter and the reviewers'comments concerning our manuscriptentitled “Optimization Design and Experimentation of Soil Covering Device for Tree Planting Machine” (agriculture-2805075). Those comments are valuable and very helpful. We haveread through comments carefully and have made corrections. Based on theinstructions provided in your letter, we uploaded the file of the revised manuscript.Revisions in the text are shown using blue highlight for additions, and strikethroughfont for deletions. The responses to the reviewer's comments are marked in red andpresented following.
We would love to thank you for allowing us to resubmit a revised copy of themanuscript and we highly appreciate your time and consideration.
Sincerely.
Xun Wu
In order to take into account the concerns of the reviewers and improve the quality of our manuscript, we have discussed and revised the issues you raised in the revised version of the manuscript.
General comments: In the simulation, the tree seedlings were not included. Do you expect the results to differ significantly with and without the presence of the tree seedlings? If so, why?
Reponse: In the process of soil covering simulation, the seedlings will not affect the simulation results. Under the condition of ensuring the soil backfill rate of the planting ditch, the seedlings will also complete the soil covering operation, which will not have a significant impact on the actual results.

Round 2
Reviewer 2 Report
Comments and Suggestions for Authors
Dear Authors,
the manuscript has been improved, but there is still a lack of discussion. If there are only a few studies available in the literature, please cite them. Use consistent citation in the appropriate position. If there are no studies available in the literature, as you suggested, please report this aspect and highlight the novelty of your work. Highlight the novelty of your model, methodology, or device in the agricultural mechanization field. This will improve the manuscript.
Author Response
Dear reviewers
Re: Manuscript ID: agriculture-2805075 and Title: Optimization Design and Experimentation of Soil Covering Device for Tree Planting Machine
Thank you for your comment on our manuscript titled "Optimization Design and Experiment of Soil Covering Device for Tree Planting Machinery" (agriculture-28005705). These opinions are very valuable and helpful. We have carefully read the comments and made corrections. According to the instructions in your letter, we have uploaded the revised manuscript file. Revisions in the text are displayed in red font for addition. The response to your opinion is as follows.
We greatly appreciate your permission to resubmit the revised copy of the manuscript, and we greatly appreciate your time and consideration.
Sincerely.
Xun Wu
We have cited literature in the introduction section of the article to explain the current research status of soil covering devices, and added references to the literature on trenching and soil covering devices for tree planting machines on the basis of the original. The specific details are as follows:
At present, there are many kinds of soil covering devices applied to agricultural machinery at home and abroad. According to the structural form, they are mainly divided into the following categories, namely single disc type, eight-shaped scraper type, double disc type and so on. Guo Hui et al. [1] designed a conical wheel-type earth-covering suppresser that integrates earth-covering and suppression functions. At the same time, the test was carried out with the thickness of the covering soil and the compactness of the soil as the test indexes, and the test results were analyzed to determine the optimal structural parameters of the conical wheel covering soil suppresser, so as to simplify the structure of the machine and improve the stability of the covering soil and suppress operation.Liao Qingxi et al. [2] designed a cigar tobacco leaf adjustable seedling bed ridging and film laying machine, which integrates soil covering operation with rotary tillage, film laying and other operation machinery. According to the operation requirements of the operation process, the structure of the soil-covered disc and the key structural parameters were designed. Guo Zhenhua et al. [3] designed an orchard rear suspension ditching and fertilizing soil covering machine, and carried out integrated design research on ditching, fertilizing and soil covering devices. By analyzing the process of soil covering operation and the effect of soil cutting and turning, a double disc soil covering mechanism with adjustable dip angle in the forward direction and vertical direction of the soil covering disc is designed.Yuanle Geng et al. [4] designed a chisel-type ditching and covering device with soil sealing function for maize no-tillage planter to solve the problems of poor soil fluidity and poor soil restoration effect in the process of maize no-tillage sowing. The theoretical analysis and design of the overburden plate are carried out, and its key parameters are determined. Ke Li [5] et al.designed a soil-covering device for sugarcane horizontal planting to solve the problem that the sugarcane soil-covering device could not meet the agronomic standards of sugarcane intercropping. Through the theoretical analysis of the covering soil device, the quadratic orthogonal rotation test and the regression analysis of the results, the key factors affecting the thickness of the covering soil were determined. Deng Xing [25] designed a sand soil continuous trenching and covering machine suitable for desertification soil in southern Xinjiang. He mainly completed the force analysis of the trenching and covering device, as well as the influence of the structural and working parameters of the trenching device on the trenching resistance. Based on the discrete element method simulation of the trenching and covering device's operation process, the optimal parameters of the trenching device were determined, but no research was conducted on the relevant parameters of the trenching and covering device.Finally, the parameters of the covering soil device were optimized.Based on the above literature and other references, it can be seen that [6-16],now, most soil covering devices are applied to crop cultivation (such as corn and cotton), and there is relatively little research on the design of soil covering devices suitable for tree planting machines. At the same time, in the research of soil covering devices, most of the current research on soil covering devices integrates soil covering with compaction, trenching and other devices. The research on soil covering devices alone is not deep enough, and most of them still design and analyze the basic structure of soil covering devices, lacking optimization design of soil covering devices.